# COMBO: Conservative Offline Model-Based Policy Optimization

**Tianhe Yu**[*,1]**, Aviral Kumar**[*,2]**, Rafael Rafailov**[1]**, Aravind Rajeswaran**[3]**,
Sergey Levine**[2]**, Chelsea Finn**[1]

[1]Stanford University, [2]UC Berkeley, [3]Facebook AI Research     ([*]Equal Contribution)

`tianheyu@cs.stanford.edu, aviralk@berkeley.edu`

## Abstract

Model-based reinforcement learning (RL) algorithms, which learn a dynamics model from logged experience and perform conservative planning under the learned model, have emerged as a promising paradigm for offline reinforcement learning (offline RL). However, practical variants of such model-based algorithms rely on explicit uncertainty quantification for incorporating conservatism. Uncertainty estimation with complex models, such as deep neural networks, can be difficult and unreliable. We empirically find that uncertainty estimation is not accurate and leads to poor performance in certain scenarios in offline model-based RL. We overcome this limitation by developing a new model-based offline RL algorithm, COMBO, that trains a value function using both the offline dataset and data generated using rollouts under the model while also additionally regularizing the value function on out-of-support state-action tuples generated via model rollouts. This results in a conservative estimate of the value function for out-of-support state-action tuples, without requiring explicit uncertainty estimation. Theoretically, we show that COMBO satisfies a policy improvement guarantee in the offline setting. Through extensive experiments, we find that COMBO attains greater performance compared to prior offline RL on problems that demand generalization to related but previously unseen tasks, and also consistently matches or outperforms prior offline RL methods on widely studied offline RL benchmarks, including image-based tasks.

## 1 Introduction

Offline reinforcement learning (offline RL) [30, 34] refers to the setting where policies are trained using static, previously collected datasets. This presents an attractive paradigm for data reuse and safe policy learning in many applications, such as healthcare [62], autonomous driving [65], robotics [25, 48], and personalized recommendation systems [59]. Recent studies have observed that RL algorithms originally developed for the online or interactive paradigm perform poorly in the offline case [14, 28, 26]. This is primarily attributed to the distribution shift that arises over the course of learning between the offline dataset and the learned policy. Thus, development of algorithms specialized for offline RL is of paramount importance to benefit from the offline data available in aforementioned applications. In this work, we develop a principled model-based offline RL algorithm that matches or exceeds the performance of prior offline RL algorithms in benchmark tasks.

A major paradigm for algorithm design in offline RL is to incorporate conservatism or regularization into online RL algorithms. Model-free offline RL algorithms [15, 28, 63, 21, 29, 27] directly incorporate conservatism into the policy or value function training and do not require learning a dynamics model. However, model-free algorithms learn only on the states in the offline dataset, which can lead to overly conservative algorithms. In contrast, model-based algorithms [26, 67] learn a pessimistic dynamics model, which in turn induces a conservative estimate of the value function. By generating and training on additional synthetic data, model-based algorithms have the potential for

35th Conference on Neural Information Processing Systems (NeurIPS 2021).

broader generalization and solving new tasks using the offline dataset [67]. However, these methods rely on some sort of strong assumption about uncertainty estimation, typically assuming access to a *model error oracle* that can estimate upper bounds on model error for any state-action tuple. In practice, such methods use more heuristic uncertainty estimation methods, which can be difficult or unreliable for complex datasets or deep network models. It then remains an open question as to whether we can formulate principled model-based offline RL algorithms with concrete theoretical guarantees on performance *without* assuming access to an uncertainty or model error oracle. In this work, we propose precisely such a method, by eschewing direct uncertainty estimation, which we argue is not necessary for offline RL.

Our main contribution is the development of conservative offline model-based policy optimization (COMBO), a new model-based algorithm for offline RL. COMBO learns a dynamics model using the offline dataset. Subsequently, it employs an actor-critic method where the value function is learned using both the offline dataset as well as synthetically generated data from the model, similar to Dyna [57] and a number of recent methods [20, 67, 7, 48]. However, in contrast to Dyna, COMBO learns a conservative critic function by penalizing the value function in state-action tuples that are not in the support of the offline dataset, obtained by simulating the learned model. We theoretically show that for any policy, the Q-function learned by COMBO is a lower-bound on the true Q-function. While

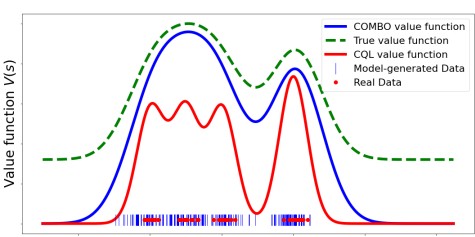

Figure 1: COMBO learns a conservative value function by utilizing both the offline dataset as well as simulated data from the model. Crucially, COMBO does not require uncertainty quantification, and the value function learned by COMBO is less conservative on the transitions seen in the dataset than CQL. This enables COMBO to steer the agent towards higher value states compared to CQL, which may steer towards more optimal states, as illustrated in the figure.

the approach of optimizing a performance lower-bound is similar in spirit to prior model-based algorithms [26, 67], COMBO crucially does not assume access to a model error or uncertainty oracle. In addition, we show theoretically that the Q-function learned by COMBO is less conservative than model-free counterparts such as CQL [29], and quantify conditions under which the this lower bound is tighter than the one derived in CQL. This is illustrated through an example in Figure 1. Following prior works [31], we show that COMBO enjoys a safe policy improvement guarantee. By interpolating model-free and model-based components, this guarantee can utilize the best of both guarantees in certain cases. Finally, in our experiments, we find that COMBO achieves the best performance on tasks that require out-of-distribution generalization and outperforms previous latent-space offline model-based RL methods on image-based robotic manipulation benchmarks. We also test COMBO on commonly studied benchmarks for offline RL and find that COMBO generally performs well on the benchmarks, achieving the highest score in 9 out of 12 MuJoCo domains from the D4RL [12] benchmark suite.

## 2 Preliminaries

**Markov Decision Processes and Offline RL.** We study RL in the framework of Markov decision processes (MDPs) specified by the tuple $\mathcal{M} = (\mathcal{S}, \mathcal{A}, T, r, \mu_0, \gamma)$. $\mathcal{S}, \mathcal{A}$ denote the state and action spaces. $T(\mathbf{s}'|\mathbf{s}, \mathbf{a})$ and $r(\mathbf{s}, \mathbf{a}) \in [-R_{\max}, R_{\max}]$ represent the dynamics and reward function respectively. $\mu_0(s)$ denotes the initial state distribution, and $\gamma \in (0, 1)$ denotes the discount factor. We denote the discounted state visitation distribution of a policy $\pi$ using $d_{\mathcal{M}}^{\pi}(\mathbf{s}) := (1 - \gamma) \sum_{t=0}^{\infty} \gamma^t \mathcal{P}(s_t = \mathbf{s}|\pi)$, where $\mathcal{P}(s_t = \mathbf{s}|\pi)$ is the probability of reaching state $\mathbf{s}$ at time $t$ by rolling out $\pi$ in $\mathcal{M}$. Similarly, we denote the state-action visitation distribution with $d_{\mathcal{M}}^{\pi}(\mathbf{s}, \mathbf{a}) := d_{\mathcal{M}}^{\pi}(\mathbf{s})\pi(\mathbf{a}|\mathbf{s})$. The goal of RL is to learn a policy that maximizes the return, or long term cumulative rewards: $\max_{\pi} J(\mathcal{M}, \pi) := \frac{1}{1-\gamma} \mathbb{E}_{(\mathbf{s}, \mathbf{a}) \sim d_{\mathcal{M}}^{\pi}(\mathbf{s}, \mathbf{a})}[r(\mathbf{s}, \mathbf{a})]$.

Offline RL is the setting where we have access only to a fixed dataset $\mathcal{D} = \{(\mathbf{s}, \mathbf{a}, r, \mathbf{s}')\}$, which consists of transition tuples from trajectories collected using a behavior policy $\pi_\beta$. In other words, the dataset $\mathcal{D}$ is sampled from $d^{\pi_\beta}(\mathbf{s}, \mathbf{a}) := d^{\pi_\beta}(\mathbf{s})\pi_\beta(\mathbf{a}|\mathbf{s})$. We define $\overline{\mathcal{M}}$ as the empirical MDP induced by the dataset $\mathcal{D}$ and $d(\mathbf{s}, \mathbf{a})$ as sampled-based version of $d^{\pi_\beta}(\mathbf{s}, \mathbf{a})$. In the offline setting, the goal is to find the best possible policy using the fixed offline dataset.

**Model-Free Offline RL Algorithms.** One class of approaches for solving MDPs involves the use of dynamic programming and actor-critic schemes [56, 5], which do not explicitly require the learning

of a dynamics model. To capture the long term behavior of a policy without a model, we define the action value function as $Q^\pi(\mathbf{s}, \mathbf{a}) := \mathbb{E}\left[\sum_{t=0}^\infty \gamma^t\, r(\mathbf{s}_t, \mathbf{a}_t) \mid \mathbf{s}_0 = \mathbf{s}, \mathbf{a}_0 = \mathbf{a}\right]$, where future actions are sampled from $\pi(\cdot|\mathbf{s})$ and state transitions happen according to the MDP dynamics. Consider the following Bellman operator: $\mathcal{B}^\pi Q(\mathbf{s}, \mathbf{a}) := r(\mathbf{s}, \mathbf{a}) + \gamma \mathbb{E}_{\mathbf{s}' \sim T(\cdot|\mathbf{s},\mathbf{a}), \mathbf{a}' \sim \pi(\cdot|\mathbf{s}')}[Q(\mathbf{s}', \mathbf{a}')]$, and its sample based counterpart: $\widehat{\mathcal{B}}^\pi Q(\mathbf{s}, \mathbf{a}) := r(\mathbf{s}, \mathbf{a}) + \gamma Q(\mathbf{s}', \mathbf{a}')$, associated with a single transition $(\mathbf{s}, \mathbf{a}, \mathbf{s}')$ and $\mathbf{a}' \sim \pi(\cdot|\mathbf{s}')$. The action-value function satisfies the Bellman consistency criterion given by $\mathcal{B}^\pi Q^\pi(\mathbf{s}, \mathbf{a}) = Q^\pi(\mathbf{s}, \mathbf{a})\ \forall(\mathbf{s}, \mathbf{a})$. When given an offline dataset $\mathcal{D}$, standard approximate dynamic programming (ADP) and actor-critic methods use this criterion to alternate between policy evaluation [40] and policy improvement. A number of prior works have observed that such a direct extension of ADP and actor-critic schemes to offline RL leads to poor results due to distribution shift over the course of learning and over-estimation bias in the $Q$ function [14, 28, 63]. To address these drawbacks, prior works have proposed a number of modifications aimed towards regularizing the policy or value function (see Section 6). In this work, we primarily focus on CQL [29], which alternates between:

**Policy Evaluation:** The $Q$ function associated with the current policy $\pi$ is approximated conservatively by repeating the following optimization:

$$Q^{k+1} \leftarrow \arg\min_Q \beta\left(\mathbb{E}_{\mathbf{s}\sim\mathcal{D}, \mathbf{a}\sim\mu(\cdot|\mathbf{s})}[Q(\mathbf{s}, \mathbf{a})] - \mathbb{E}_{\mathbf{s}, \mathbf{a}\sim\mathcal{D}}[Q(\mathbf{s}, \mathbf{a})]\right) + \frac{1}{2}\mathbb{E}_{\mathbf{s}, \mathbf{a}, \mathbf{s}'\sim\mathcal{D}}\left[\left(Q(\mathbf{s}, \mathbf{a}) - \widehat{\mathcal{B}}^\pi Q^k(\mathbf{s}, \mathbf{a})\right)^2\right], \quad (1)$$

where $\mu(\cdot|s)$ is a wide sampling distribution such as the uniform distribution over action bounds. CQL effectively penalizes the $Q$ function at states in the dataset for actions not observed in the dataset. This enables a conservative estimation of the value function for any policy [29], mitigating the challenges of over-estimation bias and distribution shift.

**Policy Improvement:** After approximating the Q function as $\hat{Q}^\pi$, the policy is improved as $\pi \leftarrow \arg\max_{\pi'} \mathbb{E}_{\mathbf{s}\sim\mathcal{D}, \mathbf{a}\sim\pi'(\cdot|\mathbf{s})}\left[\hat{Q}^\pi(\mathbf{s}, \mathbf{a})\right]$. Actor-critic methods with parameterized policies and $Q$ functions approximate $\arg\max$ and $\arg\min$ in above equations with a few gradient descent steps.

**Model-Based Offline RL Algorithms.** A second class of algorithms for solving MDPs involve the learning of the dynamics function, and using the learned model to aid policy search. Using the given dataset $\mathcal{D}$, a dynamics model $\widehat{T}$ is typically trained using maximum likelihood estimation as: $\min_{\widehat{T}} \mathbb{E}_{(\mathbf{s}, \mathbf{a}, \mathbf{s}')\sim\mathcal{D}}\left[\log \widehat{T}(\mathbf{s}'|\mathbf{s}, \mathbf{a})\right]$. A reward model $\hat{r}(\mathbf{s}, \mathbf{a})$ can also be learned similarly if it is unknown. Once a model has been learned, we can construct the learned MDP $\widehat{\mathcal{M}} = (\mathcal{S}, \mathcal{A}, \widehat{T}, \hat{r}, \mu_0, \gamma)$, which has the same state and action spaces, but uses the learned dynamics and reward function. Subsequently, any policy learning or planning algorithm can be used to recover the optimal policy in the model as $\hat{\pi} = \arg\max_\pi J(\widehat{\mathcal{M}}, \pi)$.

This straightforward approach is known to fail in the offline RL setting, both in theory and practice, due to distribution shift and model-bias [51, 26]. In order to overcome these challenges, offline model-based algorithms like MOReL [26] and MOPO [67] use uncertainty quantification to construct a lower bound for policy performance and optimize this lower bound by assuming a model error oracle $u(\mathbf{s}, \mathbf{a})$. By using an uncertainty estimation algorithm like bootstrap ensembles [43, 4, 37], we can estimate $u(\mathbf{s}, \mathbf{a})$. By constructing and optimizing such a lower bound, offline model-based RL algorithms avoid the aforementioned pitfalls like model-bias and distribution shift. While any RL or planning algorithm can be used to learn the optimal policy for $\widehat{\mathcal{M}}$, we focus specifically on MBPO [20, 57] which was used in MOPO. MBPO follows the standard structure of actor-critic algorithms, but in each iteration uses an augmented dataset $\mathcal{D} \cup \mathcal{D}_{\text{model}}$ for policy evaluation. Here, $\mathcal{D}$ is the offline dataset and $\mathcal{D}_{\text{model}}$ is a dataset obtained by simulating the current policy using the learned dynamics model. Specifically, at each iteration, MBPO performs $k$-step rollouts using $\widehat{T}$ starting from state $\mathbf{s} \in \mathcal{D}$ with a particular rollout policy $\mu(\mathbf{a}|\mathbf{s})$, adds the model-generated data to $\mathcal{D}_{\text{model}}$, and optimizes the policy with a batch of data sampled from $\mathcal{D} \cup \mathcal{D}_{\text{model}}$ where each datapoint in the batch is drawn from $\mathcal{D}$ with probability $f \in [0, 1]$ and $\mathcal{D}_{\text{model}}$ with probability $1 - f$.

## 3 Conservative Offline Model-Based Policy Optimization

The principal limitation of prior offline model-based algorithms (discussed in Section 2) is the assumption of having access to a model error oracle for uncertainty estimation and strong reliance on heuristics of quantifying the uncertainty. In practice, such heuristics could be challenging for complex datasets or deep neural network models [44]. We argue that uncertainty estimation is not

---

**Algorithm 1** COMBO: Conservative Model Based Offline Policy Optimization

---

**Require:** Offline dataset $\mathcal{D}$, rollout distribution $\mu(\cdot|\mathbf{s})$, learned dynamics model $\widehat{T}_\theta$, initialized policy and critic $\pi_\phi$ and $Q_\psi$.

1: Train the probabilistic dynamics model $\widehat{T}_\theta(\mathbf{s}', r|\mathbf{s}, \mathbf{a}) = \mathcal{N}(\mu_\theta(\mathbf{s}, \mathbf{a}), \Sigma_\theta(\mathbf{s}, \mathbf{a}))$ on $\mathcal{D}$.
2: Initialize the replay buffer $\mathcal{D}_{\text{model}} \leftarrow \varnothing$.
3: **for** $i = 1, 2, 3, \cdots,$ **do**
4:     Collect model rollouts by sampling from $\mu$ and $\widehat{T}_\theta$ starting from states in $\mathcal{D}$. Add model rollouts to $\mathcal{D}_{\text{model}}$.
5:     Conservatively evaluate $\pi_\phi^i$ by repeatedly solving eq. 2 to obtain $\hat{Q}_\psi^{\pi_\phi^i}$ using samples from $\mathcal{D} \cup \mathcal{D}_{\text{model}}$.
6:     Improve policy under state marginal of $d_f$ by solving eq. 3 to obtain $\pi_\phi^{i+1}$.
7: **end for**

---

imperative for offline model-based RL and empirically show that uncertainty estimation could be inaccurate in offline RL problems especially when generalization to unknown behaviors is required in Section 5.1.1. Our goal is to develop a model-based offline RL algorithm that enables optimizing a lower bound on the policy performance, but without requiring uncertainty quantification. We achieve this by extending conservative Q-learning [29], which does not require explicit uncertainty quantification, into the model-based setting. Our algorithm COMBO, summarized in Algorithm 1, alternates between a conservative policy evaluation step and a policy improvement step, which we outline below.

**Conservative Policy Evaluation:** Given a policy $\pi$, an offline dataset $\mathcal{D}$, and a learned model of the MDP $\hat{\mathcal{M}}$, the goal in this step is to obtain a conservative estimate of $Q^\pi$. To achieve this, we penalize the Q-values evaluated on data drawn from a particular state-action distribution that is more likely to be out-of-support while pushing up the Q-values on state-action pairs that are trustworthy, which is implemented by repeating the following recursion:

$$\hat{Q}^{k+1} \leftarrow \arg\min_Q \beta \left( \mathbb{E}_{\mathbf{s},\mathbf{a}\sim\rho(\mathbf{s},\mathbf{a})}[Q(\mathbf{s},\mathbf{a})] - \mathbb{E}_{\mathbf{s},\mathbf{a}\sim\mathcal{D}}[Q(\mathbf{s},\mathbf{a})] \right) + \frac{1}{2} \mathbb{E}_{\mathbf{s},\mathbf{a},\mathbf{s}'\sim d_f}\left[ \left( Q(\mathbf{s},\mathbf{a}) - \widehat{\mathcal{B}}^\pi \hat{Q}^k(\mathbf{s},\mathbf{a}) \right)^2 \right]. \quad (2)$$

Here, $\rho(\mathbf{s}, \mathbf{a})$ and $d_f$ are sampling distributions that we can choose. Model-based algorithms allow ample flexibility for these choices while providing the ability to control the bias introduced by these choices. For $\rho(\mathbf{s}, \mathbf{a})$, we make the following choice: $\rho(\mathbf{s}, \mathbf{a}) = d_{\hat{\mathcal{M}}}^\pi(\mathbf{s})\pi(\mathbf{a}|\mathbf{s})$, where $d_{\hat{\mathcal{M}}}^\pi(\mathbf{s})$ is the discounted marginal state distribution when executing $\pi$ in the learned model $\hat{\mathcal{M}}$. Samples from $d_{\hat{\mathcal{M}}}^\pi(\mathbf{s})$ can be obtained by rolling out $\pi$ in $\hat{\mathcal{M}}$. Similarly, $d_f$ is an $f$−interpolation between the offline dataset and synthetic rollouts from the model: $d_f^\mu(\mathbf{s}, \mathbf{a}) := f\, d(\mathbf{s}, \mathbf{a}) + (1 - f)\, d_{\hat{\mathcal{M}}}^\mu(\mathbf{s}, \mathbf{a})$, where $f \in [0, 1]$ is the ratio of the datapoints drawn from the offline dataset as defined in Section 2 and $\mu(\cdot|\mathbf{s})$ is the rollout distribution used with the model, which can be modeled as $\pi$ or a uniform distribution. To avoid notation clutter, we also denote $d_f := d_f^\mu$.

Under such choices of $\rho$ and $d_f$, we push down (or conservatively estimate) Q-values on state-action tuples from model rollouts and push up Q-values on the real state-action pairs from the offline dataset. When updating Q-values with the Bellman backup, we use a mixture of both the model-generated data and the real data, similar to Dyna [57]. Note that in comparison to CQL and other model-free algorithms, COMBO learns the Q-function over a richer set of states beyond the states in the offline dataset. This is made possible by performing rollouts under the learned dynamics model, denoted by $d_{\hat{\mathcal{M}}}^\mu(\mathbf{s}, \mathbf{a})$. We will show in Section 4 that the Q function learned by repeating the recursion in Eq. 2 provides a lower bound on the true Q function, without the need for explicit uncertainty estimation. Furthermore, we will theoretically study the advantages of using synthetic data from the learned model, and characterize the impacts of model bias.

**Policy Improvement Using a Conservative Critic:** After learning a conservative critic $\hat{Q}^\pi$, we improve the policy as:

$$\pi' \leftarrow \arg\max_\pi \mathbb{E}_{\mathbf{s}\sim\rho, \mathbf{a}\sim\pi(\cdot|\mathbf{s})}\left[ \hat{Q}^\pi(\mathbf{s}, \mathbf{a}) \right] \quad (3)$$

where $\rho(\mathbf{s})$ is the state marginal of $\rho(\mathbf{s}, \mathbf{a})$. When policies are parameterized with neural networks, we approximate the $\arg\max$ with a few steps of gradient descent. In addition, entropy regularization can also be used to prevent the policy from becoming degenerate if required [17]. In Section 4.2, we show that the resulting policy is guaranteed to improve over the behavior policy.

**Practical Implementation Details.** Our practical implementation largely follows MOPO, with the key exception that we perform conservative policy evaluation as outlined in this section, rather than using uncertainty-based reward penalties. Following MOPO, we represent the probabilistic dynamics model using a neural network, with parameters $\theta$, that produces a Gaussian distribution over the next state and reward: $\widehat{T}_\theta(\mathbf{s}_{t+1}, r|\mathbf{s}, \mathbf{a}) = \mathcal{N}(\mu_\theta(\mathbf{s}_t, \mathbf{a}_t), \Sigma_\theta(\mathbf{s}_t, \mathbf{a}_t))$. The model is trained via maximum likelihood. For conservative policy evaluation (eq. 2) and policy improvement (eq. 3), we augment $\rho$ with states sampled from the offline dataset, which shows more stable improvement in practice. It is relatively common in prior work on model-based offline RL to select various hyperparameters using online policy rollouts [67, 26, 3, 33]. However, we would like to avoid this with our method, since requiring online rollouts to tune hyperparameters contradicts the main aim of offline RL, which is to learn entirely from offline data. Therefore, *we do not use online rollouts for tuning COMBO*, and instead devise an automated rule for tuning important hyperparameters such as $\beta$ and $f$ in a fully offline manner. We search over a small discrete set of hyperparameters for each task, and use the value of the regularization term $\mathbb{E}_{\mathbf{s}, \mathbf{a} \sim \rho(\mathbf{s}, \mathbf{a})}[Q(\mathbf{s}, \mathbf{a})] - \mathbb{E}_{\mathbf{s}, \mathbf{a} \sim \mathcal{D}}[Q(\mathbf{s}, \mathbf{s})]$ (shown in Eq. 2) to pick hyperparameters in an entirely offline fashion. We select the hyperparameter setting that achieves the lowest regularization objective, which indicates that the Q-values on unseen model-predicted state-action tuples are not overestimated. Additional details about the practical implementation and the hyperparameter selection rule are provided in Appendix B.1 and Appendix B.2 respectively.

# 4 Theoretical Analysis of COMBO

In this section, we theoretically analyze our method and show that it optimizes a lower-bound on the expected return of the learned policy. This lower bound is close to the actual policy performance (modulo sampling error) when the policy's state-action marginal distribution is in support of the state-action marginal of the behavior policy and conservatively estimates the performance of a policy otherwise. By optimizing the policy against this lower bound, COMBO guarantees policy improvement beyond the behavior policy. Furthermore, we use these insights to discuss cases when COMBO is less conservative compared to model-free counterparts.

## 4.1 COMBO Optimizes a Lower Bound

We first show that training the Q-function using Eq. 2 produces a Q-function such that the expected off-policy policy improvement objective [8] computed using this learned Q-function lower-bounds its actual value. We will reuse notation for $d_f$ and $d$ from Sections 2 and 3. Assuming that the Q-function is tabular, the Q-function found by approximate dynamic programming in iteration $k$, can be obtained by differentiating Eq. 2 with respect to $Q^k$ (see App. A for details):

$$\hat{Q}^{k+1}(\mathbf{s}, \mathbf{a}) = (\widehat{\mathcal{B}}^\pi Q^k)(\mathbf{s}, \mathbf{a}) - \beta \frac{\rho(\mathbf{s}, \mathbf{a}) - d(\mathbf{s}, \mathbf{a})}{d_f(\mathbf{s}, \mathbf{a})}. \tag{4}$$

Eq. 4 effectively applies a penalty that depends on the three distributions appearing in the COMBO critic training objective (Eq. 2), of which $\rho$ and $d_f$ are free variables that we choose in practice as discussed in Section 3. For a given iteration $k$ of Eq. 4, we further define the expected penalty under $\rho(\mathbf{s}, \mathbf{a})$ as:

$$\nu(\rho, f) := \mathbb{E}_{\mathbf{s}, \mathbf{a} \sim \rho(\mathbf{s}, \mathbf{a})} \left[ \frac{\rho(\mathbf{s}, \mathbf{a}) - d(\mathbf{s}, \mathbf{a})}{d_f(\mathbf{s}, \mathbf{a})} \right]. \tag{5}$$

Next, we will show that the Q-function learned by COMBO lower-bounds the actual Q-function under the initial state distribution $\mu_0$ and any policy $\pi$. We also show that the asymptotic Q-function learned by COMBO lower-bounds the actual Q-function of any policy $\pi$ with high probability for a large enough $\beta \geq 0$, which we include in Appendix A.2. Let $\overline{\mathcal{M}}$ represent the empirical MDP which uses the empirical transition model based on raw data counts. The Bellman backups over the dataset distribution $d_f$ in Eq. 2 that we analyze is an $f$−interpolation of the backup operator in the empirical MDP (denoted by $\mathcal{B}^\pi_{\overline{\mathcal{M}}}$) and the backup operator under the learned model $\widehat{\mathcal{M}}$ (denoted by $\mathcal{B}^\pi_{\widehat{\mathcal{M}}}$). The empirical backup operator suffers from sampling error, but is unbiased in expectation, whereas the model backup operator induces bias but no sampling error. We assume that all of these backups enjoy concentration properties with concentration coefficient $C_{r,T,\delta}$, dependent on the desired confidence value $\delta$ (details in Appendix A.2). This is a standard assumption in literature [31]. Now, we state our main results below.

**Proposition 4.1.** *For large enough $\beta$, we have $\mathbb{E}_{\mathbf{s} \sim \mu_0, \mathbf{a} \sim \pi(\cdot|\mathbf{s})}[\hat{Q}^\pi(\mathbf{s}, \mathbf{a})] \leq \mathbb{E}_{\mathbf{s} \sim \mu_0, \mathbf{a} \sim \pi(\cdot|\mathbf{s})}[Q^\pi(\mathbf{s}, \mathbf{a})]$, where $\mu_0(\mathbf{s})$ is the initial state distribution. Furthermore, when $\epsilon_s$ is small, such as in the large*

*sample regime, or when the model bias $\epsilon_m$ is small, a small $\beta$ is sufficient to guarantee this condition along with an appropriate choice of $f$.*

The proof for Proposition 4.1 can be found in Appendix A.2. Finally, while Kumar et al. [29] also analyze how regularized value function training can provide lower bounds on the value function at each state in the dataset [29] (Proposition 3.1-3.2), our result shows that COMBO is less conservative in that it does not underestimate the value function at every state in the dataset like CQL (Remark 1) and might even overestimate these values. Instead COMBO penalizes Q-values at states generated via model rollouts from $\rho(\mathbf{s}, \mathbf{a})$. Note that in general, the required value of $\beta$ may be quite large similar to prior works, which typically utilize a large constant $\beta$, which may be in the form of a penalty on a regularizer [36, 29] or as constants in theoretically optimal algorithms [23, 49]. While it is challenging to argue that that either COMBO or CQL attains the tightest possible lower-bound on return, in our final result of this section, we discuss a sufficient condition for the COMBO lower-bound to be tighter than CQL.

**Proposition 4.2.** *Assuming previous notation, let $\Delta^\pi_{COMBO} := \mathbb{E}_{\mathbf{s},\mathbf{a}\sim d_{\overline{\mathcal{M}}}(\mathbf{s}),\pi(\mathbf{a}|\mathbf{s})}\left[\hat{Q}^\pi(\mathbf{s},\mathbf{a})\right]$ and*

$\Delta^\pi_{CQL} := \mathbb{E}_{\mathbf{s},\mathbf{a}\sim d_{\overline{\mathcal{M}}}(\mathbf{s}),\pi(\mathbf{a}|\mathbf{s})}\left[\hat{Q}^\pi_{CQL}(\mathbf{s},\mathbf{a})\right]$ *denote the average values on the dataset under the Q-functions learned by COMBO and CQL respectively. Then, $\Delta^\pi_{COMBO} \geq \Delta^\pi_{CQL}$, if:*

$$\mathbb{E}_{\mathbf{s},\mathbf{a}\sim\rho(\mathbf{s},\mathbf{a})}\left[\frac{\pi(\mathbf{a}|\mathbf{s})}{\pi_\beta(\mathbf{a}|\mathbf{s})}\right] - \mathbb{E}_{\mathbf{s},\mathbf{a}\sim d_{\overline{\mathcal{M}}}(\mathbf{s}),\pi(\mathbf{a}|\mathbf{s})}\left[\frac{\pi(\mathbf{a}|\mathbf{s})}{\pi_\beta(\mathbf{a}|\mathbf{s})}\right] \leq 0. \qquad (*)$$

Proposition 4.2 indicates that COMBO will be less conservative than CQL when the action probabilities under learned policy $\pi(\mathbf{a}|\mathbf{s})$ and the probabilities under the behavior policy $\pi_\beta(\mathbf{a}|\mathbf{s})$ are closer together on state-action tuples drawn from $\rho(\mathbf{s},\mathbf{a})$ (i.e., sampled from the model using the policy $\pi(\mathbf{a}|\mathbf{s})$), than they are on states from the dataset and actions from the policy, $d_{\overline{\mathcal{M}}}(\mathbf{s})\pi(\mathbf{a}|\mathbf{s})$. COMBO's objective (Eq. 2) only penalizes Q-values under $\rho(\mathbf{s},\mathbf{a})$, which, in practice, are expected to primarily consist of out-of-distribution states generated from model rollouts, and does not penalize the Q-value at states drawn from $d_{\overline{\mathcal{M}}}(\mathbf{s})$. As a result, the expression $(*)$ is likely to be negative, making COMBO less conservative than CQL.

## 4.2 Safe Policy Improvement Guarantees

Now that we have shown various aspects of the lower-bound on the Q-function induced by COMBO, we provide policy improvement guarantees for the COMBO algorithm. Formally, Proposition 4.3 discuss safe improvement guarantees over the behavior policy. building on prior work [46, 31, 29].

**Proposition 4.3** ($\zeta$-safe policy improvement). *Let $\hat{\pi}_{out}(\mathbf{a}|\mathbf{s})$ be the policy obtained by COMBO. Then, if $\beta$ is sufficiently large and $\nu(\rho^\pi, f) - \nu(\rho^\beta, f) \geq C$ for a positive constant $C$, the policy $\hat{\pi}_{out}(\mathbf{a}|\mathbf{s})$ is a $\zeta$-safe policy improvement over $\pi_\beta$ in the actual MDP $\mathcal{M}$, i.e., $J(\hat{\pi}_{out}, \mathcal{M}) \geq J(\pi_\beta, \mathcal{M}) - \zeta$, with probability at least $1 - \delta$, where $\zeta$ is given by,*

$$\mathcal{O}\left(\frac{\gamma f}{(1-\gamma)^2}\right)\underbrace{\mathbb{E}_{\mathbf{s}\sim d_{\mathcal{M}}^{\hat{\pi}_{out}}}\left[\sqrt{\frac{|\mathcal{A}|}{|\mathcal{D}(\mathbf{s})|}D_{CQL}(\hat{\pi}_{out},\pi_\beta)}\right]}_{:=\,(1)} + \mathcal{O}\left(\frac{\gamma(1-f)}{(1-\gamma)^2}\right)\underbrace{D_{TV}(\overline{\mathcal{M}},\widehat{\mathcal{M}})}_{:=\,(2)} - \underbrace{\beta\frac{C}{(1-\gamma)}}_{:=\,(3)}.$$

The complete statement (with constants and terms that grow smaller than quadratic in the horizon) and proof for Proposition 4.3 is provided in Appendix A.4. $D_{CQL}$ denotes a notion of probabilistic distance between policies [29] which we discuss further in Appendix A.4. The expression for $\zeta$ in Proposition 4.3 consists of three terms: term (1) captures the decrease in the policy performance due to limited data, and decays as the size of $\mathcal{D}$ increases. The second term (2) captures the suboptimality induced by the bias in the learned model. Finally, as we show in Appendix A.4, the third term (3) comes from $\nu(\rho^\pi, f) - \nu(\rho^\beta, f)$, which is equivalent to the improvement in policy performance as a result of running COMBO in the empirical and model MDPs. Since the learned model is trained on the dataset $\mathcal{D}$ with transitions generated from the behavior policy $\pi_\beta$, the marginal distribution $\rho^\beta(\mathbf{s}, \mathbf{a})$ is expected to be closer to $d(\mathbf{s}, \mathbf{a})$ for $\pi_\beta$ as compared to the counterpart for the learned policy, $\rho^\pi$. Thus, the assumption that $\nu(\rho^\pi, f) - \nu(\rho^\beta, f)$ is positive is reasonable, and in such cases, an appropriate (large) choice of $\beta$ will make term (3) large enough to counteract terms (1) and (2) that reduce policy performance. We discuss this elaborately in Appendix A.4 (Remark 3).

Further note that in contrast to Proposition 3.6 in Kumar et al. [29], note that our result indicates the sampling error (term (1)) is reduced (multiplied by a fraction $f$) when a near-accurate model is used to augment data for training the Q-function, and similarity, it can avoid the bias of model-based methods by relying more on the model-free component. This allows COMBO to attain the best-of-both model-free and model-based methods, via a suitable choice of the fraction $f$.

To summarize, through an appropriate choice of $f$, Proposition 4.3 guarantees safe improvement over the behavior policy without requiring access to an oracle uncertainty estimation algorithm.

## 5 Experiments

In our experiments, we aim to answer the follow questions: (1) Can COMBO generalize better than previous offline model-free and model-based approaches in a setting that requires generalization to tasks that are different from what the behavior policy solves? (2) How does COMBO compare with prior work in tasks with high-dimensional image observations? (3) How does COMBO compare to prior offline model-free and model-based methods in standard offline RL benchmarks?

To answer those questions, we compare COMBO to several prior methods. In the domains with compact state spaces, we compare with recent model-free algorithms like BEAR [28], BRAC [63], and CQL [29]; as well as MOPO [67] and MOReL [26] which are two recent model-based algorithms. In addition, we also compare with an offline version of SAC [17] (denoted as SAC-off), and behavioral cloning (BC). In high-dimensional image-based domains, which we use to answer question (3), we compare to LOMPO [48], which is a latent space offline model-based RL method that handles image inputs, latent space MBPO (denoted LMBPO), similar to Janner et al. [20] which uses the model to generate additional synthetic data, the fully offline version of SLAC [32] (denoted SLAC-off), which only uses a variational model for state representation purposes, and CQL from image inputs. To our knowledge, CQL, MOPO, and LOMPO are representative of state-of-the-art model-free and model-based offline RL methods. Hence we choose them as comparisons to COMBO. To highlight the distinction between COMBO and a naïve combination of CQL and MBPO, we perform such a comparison in Table 8 in Appendix C. For more details of our experimental set-up, comparisons, and hyperparameters, see Appendix B.

### 5.1 Results on tasks that require generalization

To answer question (1), we use two environments `halfcheetah-jump` and `ant-angle` constructed in Yu et al. [67], which requires the agent to solve a task that is different from what the behavior policy solved. In both environments, the offline dataset is

| Environment | Batch Mean | Batch Max | COMBO (Ours) | MOPO | MOReL | CQL |
|---|---|---|---|---|---|---|
| halfcheetah-jump | -1022.6 | 1808.6 | **5308.7**±575.5 | 4016.6 | 3228.7 | 741.1 |
| ant-angle | 866.7 | 2311.9 | **2776.9**±43.6 | 2530.9 | 2660.3 | 2473.4 |
| sawyer-door-close | 5% | 100% | **98.3%**±3.0% | 65.8% | 42.9% | 36.7% |

Table 1: Average returns of `halfcheetah-jump` and `ant-angle` and average success rate of `sawyer-door-close` that require out-of-distribution generalization. All results are averaged over 6 random seeds. We include the mean and max return / success rate of episodes in the batch data (under Batch Mean and Batch Max, respectively) for comparison. We also include the 95%-confidence interval for COMBO.

collected by policies trained with original reward functions of `halfcheetah` and `ant`, which reward the robots to run as fast as possible. The behavior policies are trained with SAC with 1M steps and we take the full replay buffer as the offline dataset. Following Yu et al. [67], we relabel rewards in the offline datasets to reward the halfcheetah to jump as high as possible and the ant to run to the top corner with a 30 degree angle as fast as possible. Following the same manner, we construct a third task `sawyer-door-close` based on the environment in Yu et al. [66], Rafailov et al. [48]. In this task, we collect the offline data with SAC policies trained with a sparse reward function that only gives a reward of 1 when the door is *opened* by the sawyer robot and 0 otherwise. The offline dataset is similar to the "medium-expert" dataset in the D4RL benchmark since we mix equal amounts of data collected by a fully-trained SAC policy and a partially-trained SAC policy. We relabel the reward such that it is 1 when the door is *closed* and 0 otherwise. Therefore, in these datasets, the offline RL methods must generalize beyond behaviors in the offline data in order to learn the intended behaviors. We visualize the `sawyer-door-close` environment in the right image in Figure 3 in Appendix B.4.

We present the results on the three tasks in Table 1. COMBO significantly outperforms MOPO, MOReL and CQL, two representative model-based methods and one representative model-free methods respectively, in the `halfcheetah-jump` and `sawyer-door-close` tasks, and achieves an approximately 8%, 4% and 12% improvement over MOPO, MOReL and CQL respectively on the `ant-angle` task. These results validate that COMBO achieves better generalization results in practice by behaving less conservatively than prior model-free offline methods (compare to CQL, which doesn't improve much), and does so more robustly than prior model-based offline methods (compare to MOReL and MOPO).

### 5.1.1 Empirical analysis on uncertainty estimation in offline model-based RL

To further understand why COMBO outperforms prior model-based methods in tasks that require generalization, we argue that one of the main reasons could be that uncertainty estimation is hard in these tasks where the agent is required to go further away from the data distribution. To test this intuition, we perform empirical evaluations to study whether uncertainty quantification with deep neural networks, especially in the setting of dynamics model learning, is challenging and could cause problems with uncertainty-based model-based offline RL methods such as

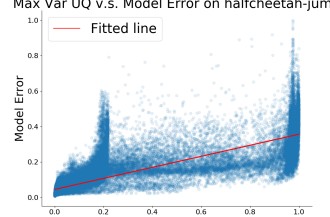 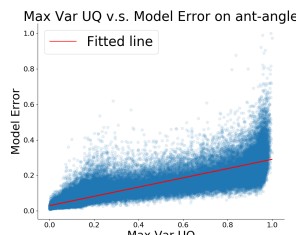

Figure 2: We visualize the fitted linear regression line between the model error and two uncertainty quantification methods maximum learned variance over the ensemble (denoted as **Max Var**) on two tasks that test the generalization abilities of offline RL algorithms (`halfcheetah-jump` and `ant-angle`). We show that **Max Var** struggles to predict the true model error. Such visualizations indicates that uncertainty quantification is challenging with deep neural networks and could lead to poor performance in model-based offline RL in settings where out-of-distribution generalization is needed. In the meantime, COMBO addresses this issue by removing the burden of performing uncertainty quantification.

MOReL [26] and MOPO [67]. In our evaluations, we consider maximum learned variance over the ensemble (denoted as **Max Var**) $\max_{i=1,...,N} \|\Sigma_\theta^i(\mathbf{s}, \mathbf{a})\|_F$ (used in MOPO).

We consider two tasks `halfcheetah-jump` and `ant-angle`. We normalize both the model error and the uncertainty estimates to be within scale $[0, 1]$ and performs linear regression that learns the mapping between the uncertainty estimates and the true model error. As shown in Figure 2, on both tasks, **Max Var** is unable to accurately predict the true model error, suggesting that uncertainty estimation used by offline model-based methods is not accurate and might be the major factor that results in its poor performance. Meanwhile, COMBO circumvents challenging uncertainty quantification problem and achieves better performances on those tasks, indicating the effectiveness and the robustness of the method.

## 5.2 Results on image-based tasks

To answer question (2), we evaluate COMBO on two image-based environments: the standard walker (`walker-walk`) task from the the DeepMind Control suite [61] and a visual door opening environment with a Sawyer robotic arm (`sawyer-door`) as used in Section 5.1.

For the walker task we construct 4 datasets: medium-replay (M-R), medium (M), medium-expert (M-E), and expert, similar to Fu et al. [12], each consisting of 200 trajectories. For `sawyer-door` task we use only the medium-expert and the expert datasets, due to the sparse reward – the agent is rewarded only when it successfully opens the door. Both environments are visulized in Figure 3 in Appendix B.4. To extend

| Dataset | Environment | COMBO (Ours) | LOMPO | LMBPO | SLAC -Off | CQL |
|---------|-------------|--------------|-------|-------|-----------|-----|
| M-R | walker_walk | **69.2** | 66.9 | 59.8 | 45.1 | 15.6 |
| M | walker_walk | 57.7 | 60.2 | **61.7** | 41.5 | 38.9 |
| M-E | walker_walk | 76.4 | **78.9** | 47.3 | 34.9 | 36.3 |
| expert | walker_walk | **61.1** | 55.6 | 13.2 | 12.6 | 43.3 |
| M-E | sawyer-door | **100.0%** | **100.0%** | 0.0% | 0.0% | 0.0% |
| expert | sawyer-door | **96.7%** | 0.0% | 0.0% | 0.0% | 0.0% |

Table 2: Results for vision experiments. For the Walker task each number is the normalized score proposed in [12] of the policy at the last iteration of training, averaged over 3 random seeds. For the Sawyer task, we report success rates over the last 100 evaluation runs of training. For the dataset, M refers to medium, M-R refers to medium-replay, and M-E refers to medium expert.

COMBO to the image-based setting, we follow Rafailov et al. [48] and train a recurrent variational model using the offline data and use train COMBO in the latent space of this model. We present

| Dataset type | Environment | BC | COMBO (Ours) | MOPO | MOReL | CQL | SAC-off | BEAR | BRAC-p | BRAC-v |
|---|---|---|---|---|---|---|---|---|---|---|
| random | halfcheetah | 2.1 | **38.8**±3.7 | 35.4 | 25.6 | 35.4 | 30.5 | 25.1 | 24.1 | 31.2 |
| random | hopper | 1.6 | 17.9±1.4 | 11.7 | **53.6** | 10.8 | 11.3 | 11.4 | 11.0 | 12.2 |
| random | walker2d | 9.8 | 7.0±3.6 | 13.6 | **37.3** | 7.0 | 4.1 | 7.3 | -0.2 | 1.9 |
| medium | halfcheetah | 36.1 | **54.2**±1.5 | 42.3 | 42.1 | 44.4 | -4.3 | 41.7 | 43.8 | 46.3 |
| medium | hopper | 29.0 | **97.2**±2.2 | 28.0 | 95.4 | 86.6 | 0.8 | 52.1 | 32.7 | 31.1 |
| medium | walker2d | 6.6 | **81.9**±2.8 | 17.8 | 77.8 | 74.5 | 0.9 | 59.1 | 77.5 | 81.1 |
| medium-replay | halfcheetah | 38.4 | **55.1**±1.0 | 53.1 | 40.2 | 46.2 | -2.4 | 38.6 | 45.4 | 47.7 |
| medium-replay | hopper | 11.8 | 89.5±1.8 | 67.5 | **93.6** | 48.6 | 3.5 | 33.7 | 0.6 | 0.6 |
| medium-replay | walker2d | 11.3 | **56.0**±8.6 | 39.0 | 49.8 | 32.6 | 1.9 | 19.2 | -0.3 | 0.9 |
| med-expert | halfcheetah | 35.8 | **90.0**±5.6 | 63.3 | 53.3 | 62.4 | 1.8 | 53.4 | 44.2 | 41.9 |
| med-expert | hopper | 111.9 | **111.1**±2.9 | 23.7 | 108.7 | 111.0 | 1.6 | 96.3 | 1.9 | 0.8 |
| med-expert | walker2d | 6.4 | **103.3**±5.6 | 44.6 | 95.6 | 98.7 | -0.1 | 40.1 | 76.9 | 81.6 |

Table 3: Results for D4RL datasets. Each number is the normalized score proposed in [12] of the policy at the last iteration of training, averaged over 6 random seeds. We take results of MOPO, MOReL and CQL from their original papers and results of other model-free methods from [12]. We include the performance of behavior cloning (**BC**) for comparison. We include the 95%-confidence interval for COMBO. We bold the highest score across all methods.

results in Table 2. On the `walker-walk` task, COMBO performs in line with LOMPO and previous methods. On the more challenging Sawyer task, COMBO matches LOMPO and achieves 100% success rate on the medium-expert dataset, and substantially outperforms all other methods on the narrow expert dataset, achieving an average success rate of 96.7%, when all other model-based and model-free methods fail.

## 5.3 Results on the D4RL tasks

Finally, to answer the question (3), we evaluate COMBO on the OpenAI Gym [6] domains in the D4RL benchmark [12], which contains three environments (halfcheetah, hopper, and walker2d) and four dataset types (random, medium, medium-replay, and medium-expert). We include the results in Table 3. The numbers of BC, SAC-off, BEAR, BRAC-P and BRAC-v are taken from the D4RL paper, while the results for MOPO, MOReL and CQL are based on their respective papers [67, 29]. COMBO achieves the best performance in 9 out of 12 settings and comparable result in 1 out of the remaining 3 settings (hopper medium-replay). As noted by Yu et al. [67] and Rafailov et al. [48], model-based offline methods are generally more performant on datasets that are collected by a wide range of policies and have diverse state-action distributions (random, medium-replay datasets) while model-free approaches do better on datasets with narrow distributions (medium, medium-expert datasets). However, in these results, COMBO generally performs well across dataset types compared to existing model-free and model-based approaches, suggesting that COMBO is robust to different dataset types.

## 6 Related Work

Offline RL [10, 50, 30, 34] is the task of learning policies from a static dataset of past interactions with the environment. It has found applications in domains including robotic manipulation [25, 38, 48, 54], NLP [21, 22] and healthcare [52, 62]. Similar to interactive RL, both model-free and model-based algorithms have been studied for offline RL, with explicit or implicit regularization of the learning algorithm playing a major role.

**Model-free offline RL.** Prior model-free offline RL algorithms have been designed to regularize the learned policy to be "close" to the behavioral policy either implicitly via regularized variants of importance sampling based algorithms [47, 58, 35, 59, 41], offline actor-critic methods [53, 45, 27, 16, 64], applying uncertainty quantification to the predictions of the Q-values [2, 28, 63, 34], and learning conservative Q-values [29, 55] or explicitly measured by direct state or action constraints [14, 36], KL divergence [21, 63, 69], Wasserstein distance, MMD [28] and auxiliary imitation loss [13]. Different from these works, COMBO uses both the offline dataset as well as model-generated data.

**Model-based offline RL.** Model-based offline RL methods [11, 9, 24, 26, 67, 39, 3, 60, 48, 33, 68] provide an alternative approach to policy learning that involves the learning of a dynamics model using techniques from supervised learning and generative modeling. Such methods however rely either on uncertainty quantification of the learned dynamics model which can be difficult for deep network models [44], or on directly constraining the policy towards the behavioral policy similar to model-free algorithms [39]. In contrast, COMBO conservatively estimates the value function by penalizing it in out-of-support states generated through model rollouts. This allows COMBO to

retain all benefits of model-based algorithms such as broad generalization, without the constraints of explicit policy regularization or uncertainty quantification.

## 7   Conclusion

In the paper, we present conservative offline model-based policy optimization (COMBO), a model-based offline RL algorithm that penalizes the Q-values evaluated on out-of-support state-action pairs. In particular, COMBO removes the need of uncertainty quantification as widely used in previous model-based offline RL works [26, 67], which can be challenging and unreliable with deep neural networks [44]. Theoretically, we show that COMBO achieves less conservative Q values compared to prior model-free offline RL methods [29] and guarantees a safe policy improvement. In our empirical study, COMBO achieves the best generalization performances in 3 tasks that require adaptation to unseen behaviors. Moreover, COMBO is able scale to vision-based tasks and outperforms or obtain comparable results in vision-based locomotion and robotic manipulation tasks. Finlly, on standard D4RL benchmark, COMBO generally performs well across dataset types compared to prior methods Despite the advantages of COMBO, there are few challenges left such as the lack of an offline hyperparameter selection scheme that can yield a uniform hyperparameter across different datasets and an automatically selected $f$ conditioned on the model error. We leave them for future work.

## Acknowledgments and Disclosure of Funding

We thank members of RAIL and IRIS for their support and feedback. This work was supported in part by ONR grants N00014-20-1-2675 and N00014-21-1-2685 as well as Intel Corporation. AK and SL are supported by the DARPA Assured Autonomy program. AR was supported by the J.P. Morgan PhD Fellowship in AI.

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
