# OpenReview forum: "COMBO: Conservative Offline Model-Based Policy Optimization"
_NeurIPS.cc/2021/Conference — NeurIPS 2021 Poster_

### Official Review · Reviewer_nBvp · 2021-07-05

**Rating:** 8
**Confidence:** 4

**Summary:**

The paper presents a new approach for model-based offline RL. Here, the model is not used to estimate the return directly using roll-outs, but to generate synthetic data that is then used to train the Q-function. In a way, the method is a combination of CQL and MOPO. What is new is the idea of using this synthetic data to, penalize extrapolation instead of using uncertainty estimation.
This modification performs much better on the selected benchmarks.
As benchmarks a selection of D4RL benchmarks is used (3 benchmarks) and two vision based benchmarks.


**Limitations And Societal Impact:**

Yes

**Main Review:**

Originality:\
Very good.
On the one hand, it is a combination of existing methods (CQL and MOPO), but on the other hand, the idea of using the generated synthetic data to penalize the extrapolation seems to me to be a significant idea. Simple, elegant, but seemingly functional.

Quality:\
High quality.

Clarity:\
Very clear.

Significance:\
I think the idea to use the synthetic data as means for behavior regularization might be very influential.

Further remarks:\
The results would be even more convincing if the experiments had been performed with more than just three trials.

**Update after feedback:** \
Based on the feedback, discussion, and very competent responses and explanations from the authors, I have raised my score to 8.

The present results show statistically significantly that COMBO performs best in the experiments conducted.
A simple t-test (student-t with 5 degrees of freedom) or a rank test clearly show this.

Furthermore, I think it is clear that the algorithm is not plagiarism, but a new development, even if it is essentially a combination of two known methods.
My opinion is that it is absolutely common to create something new by combining what is known. The fact that this combined algorithm works recognizably better is an interesting finding worth presenting to the specialized public.

Regarding the treatment of hyperparameters, I want to state my conviction from many years of practice that on the one hand the best comparison between different methods is the one where each method is near 1) its respective optimum in the hyperparameter space (otherwise one compares the suitability of the hyperparameters rather than the methods) and that in the applications of offline RL the data efficiency and performance were always in the foreground and not the effort for finding suitable hyperparameters. In short, if it is POSSIBLE, it will be done. On this topic, I find the authors' response remarkably thoughtful and coherent.

1\) by spending the same effort to come near the respective optimum in hyper-parameter space for each method, if the difficulty of finding the optimum differs significantly between the different methods






**Time Spent Reviewing:**

11

---

> ### Author Response · Authors · 2021-08-09
> **Author Response**
>
> Thank you for your review and for a positive assessment of our work! We respond to the concerns below:
>
> **The results would be even more convincing if the experiments had been performed with more than just three trials.** To address the comment, we are now updating the paper to include **six** total seeds for COMBO and the second-best performing method on all the tasks. Below, we present results with six seeds on tasks that require generalization (Table 1) for COMBO and MOPO (the second-best performing method on these generalization tasks), that have finished running. Here are the results ($\pm$ denotes the 95%-confidence interval of the various runs):
>
> |Task             |Batch mean|Batch Max|COMBO (ours)     |MOPO            |
> |-----------------|----------|---------|-----------------|----------------|
> |halfcheetah-jump | -1022.6  | 1808.6  | 5308.7$\pm$575.5| 4053.4$\pm$67.0|
> |ant-angle        | 866.7    | 2311.9  | 2776.9$\pm$43.6 | 809.2$\pm$73.0 |
> |sawyer-door-close| 5%       | 100%    | 98.3$\pm$3.0%   | 66.3$\pm$37.5% |
>
> As shown in the table above, the performance of COMBO is stable across random seeds and consistently outperforms the second-best approach, MOPO. We will run six random seeds of COMBO and other baselines on all the other domains and report 95%-confidence intervals in the final version of the paper.

---

> > ### Comment · Reviewer_nBvp · 2021-08-17
> > **No further concerns**
> >
> > I am glad that the authors now present their results on the statistics of 6 trials and have no further concerns.

---

### Official Review · Reviewer_CcvH · 2021-07-16

**Rating:** 6
**Confidence:** 4

**Summary:**

This paper proposed a new model-based offline RL algorithm, COMBO. The algorithm applies model-based learning to be able to generate random rollouts, and the generated data is combined with the given fixed dataset for value function training. The paper shows by using the generated rollouts as data for Conservative Q-learning provides better constraint and better empirical performance compared to the orignal CQL algorithm.

**Limitations And Societal Impact:**

Yes!

**Main Review:**

This paper proposed algorithm COMBO, which is a combination of model-based RL and offline RL algorithm CQL. Instead of using random generation to explore (s, a) space for Q value constraining, the algorithm used model-based learning to learn the environment dynamics and then applied auxiliary policy to generate random rollouts for the use of constraining Q value. Theoretical proof is provided to show that the new Q update is conservative and the policy improvement is guaranteed. There are no new ideas proposed, but combining these two and making the algorithm work in most scenarios are precious work. Extensvie experiments are done to show COMBO performs well in generalization tasks, image-based tasks and OpenAI Gym tasks.

The papar is well written and well structured. After reading, I have the following questions for the authors:
1. Do you have a ablation study of using which kind of rollout policy \mu(a|s) to generate addtional data points is the best for improving the performance?

2. Experiment details: for D4RL experiments, do you use the datasets in version 'v0' or 'v2' for performance comparison?

3. It seems that the hyperparameter tuning is based on each single task, which is not a standard way for performance comparison in RL? Are the table of results shown in the paper based on the same set of hyperparameters or several set of hyperparameters? There is no validation dataset for RL task. If you are tunning hyperparameters on real rollouts, that is a really dangerous thing in practice, for example in medical and financial areas.

Minor:

In proposition 4.2, the \delta_{COMBO} has the same (s, a) input distribution as \delta_{CQL}, which I think it should be different based on your notations in the paper.

In proposition 4.1, Equation (*), I think the subscript \pi(a|s) is not needed in the second term.

Line 68: under which the this?


**Time Spent Reviewing:**

6

---

> ### Author Response · Authors · 2021-08-09
> **Author Response**
>
> Thank you for your review! To address the comment regarding the idea, we have added an experiment showing the importance of penalizing the Q-value on model-generated states in COMBO, which a naive combination of existing ideas (CQL and MBPO) does not do; we have added the requested ablations for the model rollout policy; added a discussion of hyperparameter tuning and addressed the minor concerns. We would appreciate it if you point out any remaining concerns and we will be happy to respond to them. We detail our response below:
>
> **"There are no new ideas proposed, but combining these two and making the algorithm work in most scenarios are precious work."**
>
> We appreciate your positive feedback. However, we would like to clarify that while COMBO utilizes Q-value regularization similar to CQL, the effect is very different. CQL only penalizes the Q-value on unseen actions on the states observed in the dataset whereas COMBO penalizes Q-values on states generated by the learned model while maximizing Q values on state-action tuples in the dataset.
> To empirically demonstrate the consequences of this distinction, naively applying CQL to the combination of offline data and model generated data (CQL + MBPO) performs quite a bit worse than COMBO on generalization experiments (paper Sec 5.1) as shown below. The results are averaged across 6 random seeds ($\pm$ denotes the 95%-confidence interval of the various runs). This suggests that carefully considering the state distribution, as done in COMBO, is crucial.
>
> |Task             |Batch mean|Batch Max|COMBO (ours)     |CQL+MBPO        |
> |-----------------|----------|---------|-----------------|----------------|
> |halfcheetah-jump | -1022.6  | 1808.6  | **5308.7**$\pm$575.5| 4053.4$\pm$176.9|
> |Ant-angle        | 866.7    | 2311.9  | **2776.9**$\pm$43.6 | 809.2$\pm$135.4|
>
> We will add this comparison and a discussion to the revised paper.
>
> **"It seems that the hyperparameter tuning is based on each single task, which is not a standard way for performance comparison in RL? If you are tunning hyperparameters on real rollouts, that is a really dangerous thing in practice."**
>
> Prior work on model-based offline RL generally _does_ select hyperparameters per task (see, e.g., MOPO, MOReL, MBOP, etc). However, we agree that this is not good practice. On the other hand, hyperparameter tuning for offline RL methods is an open question (see Levine et al. 2020 for a discussion). Even off-policy evaluation methods, explicitly built with the goal of addressing hyperparameter tuning and policy selection, often fail to estimate values of policies reliably, rendering BC as the most optimal method (see Qin et al. 2021 for empirical evidence). Therefore, to be able to select some hyperparameters, we devised an automated rule for tuning certain hyperparameters for COMBO without requiring manual selection per task.
>
> Our automated procedure goes as follows: we search over a small discrete set of hyperparameters for each task. We do not use real rollouts for this search, but instead use the value of the regularization term $\mathbb{E}\_{\mathbf{s}, \mathbf{a} \sim \rho(\mathbf{s},\mathbf{a})}\left[Q(\mathbf{s},\mathbf{a})\right]-\mathbb{E}\_{\mathbf{s}, \mathbf{a} \sim \mathcal{D}}\left[Q(\mathbf{s},\mathbf{a})\right]$ (shown in Eq.2 in the paper)  to pick hyperparameters in a completely offline fashion. We pick the hyperparameter that achieves the lowest regularization objective, which indicates that the Q-values on unseen model-predicted state-action tuples are not overestimated. Since we perform hyperparameter selection in the fully offline setting without requiring any rollouts in the real environment, we believe that it is fine to pick hyperparameters for each task. We will add this offline hyperparameter selection scheme in the final version of the paper. A principled method for selecting hyperparameters for COMBO, like any other offline RL algorithm, is an open question and we have added this as an avenue for future work.
>
> Qin et al. NeoRL: A Near Real-World Benchmark for Offline Reinforcement Learning, arXiv 2102.00714
>
> **"Do you have an ablation study of using which kind of rollout policy \mu(a|s) to generate additional data points is the best for improving the performance?"**
>
> We have now performed the requested ablation study on the tasks that require generalization with 6 random seeds and will perform this ablation study on all the other tasks for the final. We choose two rollout policies, a uniform-at-random policy, $\text{Unif}(a)$, and the learned policy, $\pi(a|s)$. As shown in the results below, we find that using $\pi(a|s)$ achieves the best performance on halfcheetah-jump and sawyer-door-close while using the random policy performs better on ant-angle. We used the learned policy as the rollout policy for our results since it led to a lower value of the regularizer, consistent with the hyperparameter selection scheme as discussed in the previous paragraph.
>
> |Task             |Batch mean|Batch Max|COMBO (with $\pi(a&#124;s)$)|COMBO (with $\text{Unif}(a)$)|
> |-----------------|----------|---------|-----------------------|-----------------------------|
> |halfcheetah-jump | -1022.6  | 1808.6  | **5308.7**$\pm$575.5      | 1221.0$\pm$44.5             |
> |ant-angle        | 866.7    | 2311.9  | 921.2$\pm$605.1       | **2776.9**$\pm$43.6             |
> |sawyer-door-close| 5%       | 100%    | **98.3**$\pm$3.0%         | 45.0$\pm$22.5%              |
>
>
> **"For D4RL experiments, do you use the datasets in version 'v0' or 'v2' for performance comparison?"**
>
> Following previous offline RL works (CQL, MOPO, MOReL, etc.) and to facilitate direct comparison against numbers from prior work, we use version ‘v0’ for D4RL experiments. We will clarify this in the paper.
>
> **Minor points:** Thank you for pointing out the minor clarity concerns and typos. We will address these in the revised version of the paper.
>
> We are happy to address any more concerns that are remaining.

---

> ### Author Response · Authors · 2021-08-16
> **Discussion**
>
> Thank you again for your review! Please let us know if our response below addresses your concerns about the paper. We will be happy to clarify any other concerns/questions that may be remaining.

---

> > ### Comment · Reviewer_CcvH · 2021-08-23
> > **Discussion**
> >
> > Thanks for your repsonse and additional experiments! However, I think some of my concerns are still there.
> >
> > My first concern is that the work COMBO is simply a combination of CQL and MBPO.  The ablation experiment does not make sense to me. You provide an additional experiment which "naively applys CQL to the combination of offline data and model generated data (CQL + MBPO)" and you show that the CQL "performs quite a bit worse than COMBO". This definitely is not going to work well, especially for offline RL, since now it is applying CQL onto new dataset generated by MBPO, which will surely introduces more OOD issues. So, I think the main contribution of COMBO is using model generated data (by MBPO) instead of using random generated data for the CQL constraint term + using MBPO for the training of actor, which is a combination of CQL and MBPO from my perspective.
> >
> > My second concern is still the per-task hyperparameter tuning. Per-task hyperparameter tuning is making things complicated and is kind of unfair with other algorithms, which shows results with the uniform set of hyperparameters. There are a lot of compliants in CQL github, that their results can not be successfully reproduced, which I think it could be possiblly due to the per-task hyperparameter tuning. As for your proposed automated tuning method, I am surprised that you didn't mention anything about the tuning method in your submission (both main and appendix). If per-task hyperparameter tuning is necessay for producing good performance, it is a sign that the algorithm itself cannot work generally well. However, the generality across tasks are important for real world data. Currently, all the tasks being evaluated are still simulated data (from simulators like MuJoCo), while real world data could be more complicated where algorithm with high sensitivity to hyperparameter tuning could possibly not work. For example, there are conplaints about the real performance of DDPG, because of its sensitivity to hyperparameter tuning.  Moreover, COMBO relies on per-task hyperparameter for several parameters, where one or two is the limit of my acceptance. You didn't provide a whole table about what the actual hyperparameters that you used for all tasks, instead you vaguely gave a range of possible values, for example "{1e −4, 3e − 4} for the Q-function learning rate", which will prohibit the continuing work to reproduce the results.

---

> > > ### Author Response · Authors · 2021-08-25
> > > **Author Response to the Discussion Questions: Clarification to COMBO vs CQL + MBPO (1/2)**
> > >
> > > Thank you for your reply! We answer specific questions as follows in two threads.
> > >
> > > # 1. COMBO vs CQL + MBPO
> > >
> > > We would like to clarify the distinction between: (1) naively combining CQL and MBPO, and COMBO, and (2) the combination of CQL and MBPO proposed by the reviewer and COMBO.
> > >
> > >
> > > **Naive CQL + MBPO vs COMBO:** While a naive application of the CQL regularizer that handles OOD actions will prescribe minimizing the Q-values on unseen actions and maximizing them on actions from the behavior policy’s action distribution, regardless of the state distribution, the COMBO regularizer consists of terms that penalize Q-values and maximize Q-values on different state distributions. COMBO minimizes Q-values on states sampled from model rollouts and actions drawn randomly, while maximizing only the Q-values only on states from the dataset.
> > >
> > >
> > > **Reviewer’s suggested combination of CQL + MBPO vs COMBO:** Please note that COMBO uses the MBPO-style training (i.e., model-generated data) for both the Bellman update and the policy update, which is crucial and is different from what is referred to as “a combination of CQL and MBPO” from the reviewer’s perspective.
> > >
> > > **To demonstrate the significance of COMBO empirically**, we implemented the method of combining CQL and MBPO suggested by the reviewer: “using model data for CQL constraint + MBPO training for the actor” and found that it performs significantly worse than COMBO as shown in the table below. The performance gap between this method and COMBO, is because using the model data for Bellman update allows the Q-function to be less conservative, as discussed in Remark 1 (Appendix A.1), since  f < 1. On the other hand, not using the model data for the Bellman update will make the learned Q-values very conservative, which is expected to give rise to worse performance. We would also point out that the naive combination of CQL + MBPO presented in our previous response and also shown below works better than model-free CQL on one of the tasks and much better than the combination of CQL and MBPO suggested by the reviewer above, and so we are not sure why this is “definitely not going to work well”. Overall, these experiments show that COMBO is better than at least two different ways of naively combining CQL and MBPO, and is thus both not obvious and involves decisions that are important for good performance.
> > >
> > > |Task             |Batch mean|Batch Max|COMBO (ours)     | Reviewer CcvH suggested combination of CQL and MBPO |CQL+MBPO        |
> > > |-----------------|----------|---------|-----------------|----------------|----------------|
> > > |halfcheetah-jump | -1022.6  | 1808.6  | **5308.7**$\pm$575.5| -1914.6$\pm$143.7 | 4053.4$\pm$176.9|
> > > |Ant-angle        | 866.7    | 2311.9  | **2776.9**$\pm$43.6 | -38.1$\pm$241.6 | 809.2$\pm$135.4|
> > >
> > > **To summarize**, we believe that these method differences are significant. The obvious way of combining CQL and MBPO, and the method suggested by the reviewer, do not work well. COMBO is simple, yet not obvious. The contribution of our paper is not to just heuristically combine these methods: we include a thorough theoretical analysis, and our empirical results show that this simple approach results in significant gains in performance.
> > >
> > > In the same sense, one could view previously published, highly-cited and impactful works such as TD3 (Fujimoto et al. 2019) as combining DDPG (Lillicrap et al. 2015) with double Q-learning (Hasselt et al. 2010), or SAC (Haarnoja et al. 2019) as combining soft Q-learning (Haarnoja et al. 2018) with an actor-critic formulation, or TRPO (Schulman et al. 2015) as combining neural networks with natural policy gradient (Kakade, 2002), but this doesn’t mean that these algorithms are not significant due to their empirical and/or theoretical contributions. In fact, these algorithms have served as building blocks of many future works.

---

> > > > ### Author Response · Authors · 2021-08-25
> > > > **Author Response to the Discussion Questions: Addressing concern on Per-Task Hyperparameter Tuning (2/2)**
> > > >
> > > > # 2. Concern on Per-Task Hyperparameter Tuning
> > > >
> > > > First, we want to clarify that the automated selection rule mentioned in our previous response is applied _completely offline_, with no access to online rollouts, and removes the need to tune hyperparameters per-task or accessing online rollouts to measure policy performance. We believe this addresses the issue you raised, but if not, please let us know. This is similar to how automated selection rules can be used to obtain values for hyperparameters in supervised learning -- for example, how the value of the validation loss computed on a held-out set in supervised learning can be used to decide reasonable values of certain hyperparameters without peeking at the test accuracy.
> > > >
> > > > More generally, we would note that it is unreasonable to expect identical values of hyperparameters to work well on all tasks; this is not even the case in supervised learning. For example, hyperparameters for different pairs of languages in neural machine translation (e.g. En-De and En-Fr), or two different image classification problems (e.g., Imagenet and CIFAR-10) are different. Similarly, expecting similar hyperparameters to work on simulated MuJoCo locomotion tasks and a real autonomous driving task is unreasonable for any algorithm, be it supervised learning or reinforcement learning. However, the crucial piece that still makes supervised learning general is an automated rule for obtaining the values of hyperparameters. Similarly, our automated selection rule for the values of parameters based on the COMBO regularizer alleviates the need for doing any manual per-task hyperparameter tuning by peeking at the learned policy performance via online rollouts.
> > > >
> > > > **Why COMBO is a fair comparison against other algorithms:** Note that prior model-based RL schemes like MOPO and MOReL have similar hyperparameters as COMBO and tune them manually based on policy performance obtained via online rollouts. On the other hand, the automated selection rule in COMBO returns values of these hyperparameters without needing to access online rollouts, which would be unavailable on any real task where we would want to apply offline RL. Note that the total number of training runs we need to run is of similar magnitude for COMBO and prior methods, but COMBO doesn’t need peeking at the learned policy performance via online rollouts, while prior methods do require this, including those methods that utilize a uniform set of hyperparameters. This indicates that in a similar number of hyperparameter runs, COMBO should be preferred because it attains good performance without needing online rollouts, which is a plus point of COMBO.
> > > >
> > > > We do agree with the reviewer that a method equipped with an automated selection rule that returns identical hyperparameter values for all tasks in D4RL without requiring online rollouts to do so is more desirable, but we are unaware of any recent offline RL method, model-free or model-based, that has been shown to have this capability. Even off-policy evaluation methods designed specifically for hyperparameter selection perform poorly on D4RL domains (see Figure 5 in Fu et al. ICLR 2021, “Benchmarks for Deep Off-Policy Evaluation”). The next best alternative is to let the automated selection rule output different hyperparameters for each task, but without accessing online rollouts or manual hyperparameter tuning, which is what COMBO does.
> > > >
> > > > **Results and modifications we will make to the paper:** We agree with the reviewer that the automatic hyperparameter selection rule is an important component that needs to be discussed at length in the paper as a part of our proposed method and we apologize for not clearly stating it in the submission. We will add this very explicitly to the final version of the paper. Below, we provide empirical evidence on representative D4RL domains showing how our automated selection rule that selects hyperparameters with the smallest value of the COMBO regularizer is effective in obtaining a policy with a good performance, without using any online rollouts. We will include the complete table in the final version of the paper.
> > > >
> > > > |Task             |$\beta=0.5$|$\beta=0.5$            |$\beta=5.0$|$\beta=5.0$            |
> > > > |-----------------|----------|---------|---------|---------|
> > > > |             |performance|regularizer value           |performance|regularizer value           |
> > > > |halfcheetah-medium        |  **54.2**  | **-778.6**  | 40.8  | -236.8  |
> > > > |halfcheetah-medium-replay| **55.1**       | **28.9**    |9.3       | 283.9    |
> > > > |halfcheetah-medium-expert| 89.4       | 189.8    | **90.0**       | **6.5**    |
> > > > |hopper-medium        |  75.0  | -740.7  | **97.2**  | **-2035.9** |
> > > > |hopper-medium-replay| **89.5**       | **37.7**    |28.3       | 107.2    |
> > > > |hopper-medium-expert| **111.1**       | **-705.6**    |75.3       | -64.1    |
> > > > |walker2d-medium        |  1.9  | 51.5  | **81.9**  | **-1991.2**  |
> > > > |walker2d-medium-replay| **56.0**       | **-157.9**    |27.0       | 53.6    |
> > > > |walker2d-medium-expert| 10.3       | -788.3    |**103.3**       | **-3891.4**    |
> > > >
> > > >
> > > > | Task            | $\mu(a\|s)=\text{Unif}(a)$ | $\mu(a\|s)=\text{Unif}(a)$ | $\mu(a\|s)=\pi(a\|s)$ | $\mu(a\|s)=\pi(a\|s)$ |
> > > > |-----------------|----------------------------|----------------------------|-----------------------|-----------------------|
> > > > |                 | performance                | regularizer value          | performance           | regularizer value     |
> > > > | hopper-medium   | **97.2**                   | **-2035.9**                | 52.6                  | -14.9                 |
> > > > | walker2d-medium | 7.9                        | -106.8                     | **81.9**              | **-1991.2**           |
> > > >
> > > > |Task             |$\rho(s) = d^\pi_{\hat{\mathcal{M}}} $|$\rho(s) = d^\pi_{\hat{\mathcal{M}}}$            |$\rho(s) = d_f$|$\rho(s) = d_f$          |
> > > > |-----------------|----------|---------|---------|---------|
> > > > |             |performance|regularizer value           |performance|regularizer value           |
> > > > |hopper-medium        | **97.2**  | **-2035.9** |  56.0  | -6.0  |
> > > > |walker2d-medium        |  1.8  | 14617.4  | **81.9**  | **-1991.2**  |
> > > >
> > > > |Task             |$f = 0.5 $|$f = 0.5$            |$f = 0.8$|$f = 0.8$          |
> > > > |-----------------|----------|---------|---------|---------|
> > > > |             |performance|regularizer value           |performance|regularizer value           |
> > > > |hopper-medium        | **97.2**  | **-2035.9** |  93.8  | -21.3  |
> > > > |walker2d-medium        |  70.9  | -1707.0  | **81.9**  | **-1991.2**  |
> > > >
> > > > These results indicate that for all the four hyperparameters considered -- **(1)** the value of $\beta$ in the COMBO regularizer, **(2)** the choice of policy $\mu(a|s)$ for model rollouts, **(3)** the choice of $\rho(s, a)$ under which Q-values are penalized, and **(4)** the choice of the ratio between model rollouts and offline data, automatically selecting the hyperparameter which attains the smallest value of the regularizer gives rise to the highest policy performance, indicating the effectiveness of our automatic selection scheme.
> > > >
> > > > We also agree with the reviewer that we didn’t present the hyperparameters for each task in a table. However, we would like to note that we have the exact hyperparameters in Appendix B.2, just mentioned in words, which we will cleanly present in a tabular form in the revised version.

---

> > > > > ### Comment · Reviewer_CcvH · 2021-08-26
> > > > > **Response**
> > > > >
> > > > > Thanks for providing more details! I do agree that the automated selection rule applied completely offline is a good way to do hyparameter selection. I appreciate you will share all the detailed hyperparameter values in table in a tabular form in the revised version. I am going to raise my score to 6.

---

> > > > > > ### Author Response · Authors · 2021-08-26
> > > > > > **Thanks for raising the score**
> > > > > >
> > > > > > Thank you for raising the score! We are glad that the responses and revisions have addressed your concerns.

---

> > > > ### Comment · Reviewer_CcvH · 2021-08-26
> > > > **Response**
> > > >
> > > > Thanks for you response again!
> > > >
> > > > I am sorry that I should be more elaborate on my wordings, and I am sorry that make you misunderstand my sentence "using model data for CQL constraint", where the “constraint" means the constraint term, which means minimizing the Q_value estimates on the new generated data, since they are all unseen data compared to the given dataset. Using the generated data for Bellman update (you did experiment), I agree it will make the performance much worse.
> > > >
> > > > Thanks for the detailed comparison. I agree that COMBO has its distinctions. COMBO used a mixed and ratio modified dataset (f * given dataset +  (1 - f) * generated data) for doing Bellman update, while uses only the generated states and randomly sampled actions for minimizing Q and maximizes Q value on state-action pairs in given dataset. COMBO uses the MBPO-style training for both the Bellman update and the policy update.

---

### Official Review · Reviewer_4HWY · 2021-07-18

**Rating:** 7
**Confidence:** 2

**Summary:**

This work investigates model-based reinforcement learning with offline dataset. In particular, most of the work on mbrl with offline datasets requires uncertainty estimation to determine the out-of-distribution states and actions. This work eliminates the need to have uncertainty estimation by using the existing Conservative Q-Learning(CQL) algorithm for states and actions from the dataset as well as the rollouts generated from the dynamics using the policy being learned. They show that inclusion of new dataset from the generated rollouts and by being pessimistic about their values, they tend to be better than existing Model-based offline Reinforcement Learning which requires uncertainty estimation

**Ethics Review Area:**

["I don’t know"]

**Limitations And Societal Impact:**

Yes, the authors talk about limitations of their work in terms of potential miss-reward specification which could be harmful for real-world. This especially arises due to learning of parametric dynamics models

**Main Review:**

Originality and Significance:  Uncertainty estimation for model based offline reinforcement learning has been required for determining out of distribution state and actions. This is done either using an oracle estimator or by learning an ensemble of models(dynamics) and measuring the divergence between the states generated by these ensembles. This work eliminates the need to have this uncertainty metric by simply minimizing the value estimates for all states and actions out of the given dataset; as done in the CQL algorithm. They supplement CQL by addition of new data generated using the model(dynamics) and they treat the  states and actions in the generated rollout as out of distribution.

Quality:  They support their approach via a theoretical analysis to indicate the learned critic estimates would be lower-bounded than the actual values given by the bellman operator with the oracle transition model. Irrespective of inaccuracies in the learned model which can give overestimated reward to states generated in rollouts, they simply depend on the \beta in eq. 2 for suppression of Q-values. I am wondering, how do they choose this value in practice?  And, did this choice affected their results for random-hopper? And medium-reply hopper in table 3.

They experimented with both image-based environments as well as the d4rl dataset. It was not clear to me if they were using latent dynamics or observation-based dynamics for the image-based experiments.

Clarity: Yes, the ideas are conveyed in a clear manner. And, the given equations are enough for implementation; given the proposed method would only require a minimal change in existing open-source implementations of the baseline algorithms.


**Time Spent Reviewing:**

5

---

> ### Author Response · Authors · 2021-08-09
> **Author Response**
>
> Thank you for your review and a positive assessment of our paper! We answer specific questions as follows:
>
> **"How do they choose this value in practice? And, did this choice affect results for random-hopper? And medium-reply hopper in table 3."**
>
> As discussed in Appendix B.2, we select $\beta$ by searching over the set of {0.5, 1.0, 5.0}. Since we do not have access to online rollouts to select $\beta$ (as we operate in an offline setting), we instead use the value of the regularization term
> $\\mathbb{E}\_{\\mathbf{s}, \\mathbf{a} \\sim \\rho(\\mathbf{s},\\mathbf{a})}[Q(\\mathbf{s},\\mathbf{a})]-\\mathbb{E}\_{\\mathbf{s}, \\mathbf{a} \\sim \\mathcal{D}}[Q(\\mathbf{s},\\mathbf{a})]$
> (shown in Eq.2 in the paper) to determine the choice $\beta$. We always pick \beta with the lowest value of this regularization term, since that corresponds to Q functions that are sufficiently regularized to not erroneously overestimate the Q-values on states visited in model-produced rollouts.
> Regarding the hyperparameter choice on hopper-random and hopper-medium-replay, we follow the hyperparameter selection scheme described above and find that $\beta=1.0$ is selected by the aforementioned selection scheme. We hypothesize that in this domain $\beta=0.5$ is likely too small to prevent over-estimated Q-values whereas $\beta=5.0$ is likely too large and hence leads to overly underestimated Q-functions. We will add this information to the revised paper.
>
> **"It was not clear to me if they were using latent dynamics or observation-based dynamics for the image-based experiments."**
>
> As mentioned in L365-366, we follow prior work [45] and use a latent dynamics model. We will further clarify this in the final version of the paper.

---

### Official Review · Reviewer_GPHd · 2021-07-21

**Rating:** 6
**Confidence:** 4

**Summary:**

This paper introduces a Conservative Offline Model-Based policy Optimization RL algorithm, called COMBO, that learns a pessimistic model by enabling a lower bound optimization on the policy performance without requiring uncertainty quantification. COMBO employs an actor-critic method to learn the value function on both offline and synthetic datasets. The idea is to penalize Q-values learned from state-action pairs that are out-of-support of the offline dataset.


**Limitations And Societal Impact:**

The authors adequately addressed the limitations and future directions in the conclusion of the paper.

**Main Review:**

$\textbf {Overview}$:

This paper proposes a new technique for conservative offline model-based policy optimization. Theoretical guarantees are provided to demonstrate that: 1) COMBO achieves less conservative Q-values compared to prior model-free offline RL methods, 2) COMBO's learned Q-function is the lower bound of the true Q-function that eliminates the need to access a model error or uncertainty oracle in model-based RL, and 3) it guarantees safe policy improvement. Empirical results show the outstanding generalization performance of COMBO and its adaptability to unseen behaviors. Moreover, it can be applied in vision-based tasks and achieve comparable results.

$\textbf {Major concerns}$:

- Despite the outstanding performance of the COMBO in most of the standard benchmarks compared to the state-of-the-art model-based and model-free algorithms, and the theoretical guarantees discussed in the paper, COMBO is still dependent on the sub-optimality of the data (offline + synthetic). This issue can also be inferred from the occasional inferiority of COMBO comparing to model-based methods when dealing with more diverse state-action distributions.

- To evaluate how well COMBO generalizes to the new tasks, experimental results are averaged over only three random seeds. This can introduce some bias in the final results due to the low number of runs. Lack of insight into the standard deviation values and the inadequate number of independent runs damage the reliability of comparing different methods.


$\textbf {Minor concern}$:

- Regarding the results demonstrated in D4RL tasks, it is not adequately explained how they have normalized return values of COMBO and other methods neither in the paper nor in the appendix. I would like the authors to explain how they have chosen the expert scores for every three environments? How did they fine-tune the trained policy with behavioral cloning [1]?

- Since source code availability is not mentioned in the paper (but included in supplementary), I would invite the authors to release their code.

Overall, I found the paper is well written, the proposed technique interesting, and the motivation clear. To the best of my knowledge, what is proposed in this article is novel.

[1] Justin Fu, Aviral Kumar, Ofir Nachum, George Tucker, and Sergey Levine. D4rl: Datasets for deep data-driven reinforcement learning, 2020.

**Time Spent Reviewing:**

15

---

> ### Author Response · Authors · 2021-08-09
> **Author Response**
>
> Thank you for your review and for a positive assessment of the paper! To address the concern regarding the statistical significance of the results, we are now updating the paper to include **six** total seeds for COMBO and the second-best performing method on all the tasks. Below, we present results with six seeds on tasks that require generalization (Table 1) for COMBO and MOPO (the second-best performing method on these generalization tasks), that have finished running. Here are the results ($\pm$ denotes the 95%-confidence interval of the various runs):
>
> |Task             |Batch mean|Batch Max|COMBO (ours)     |MOPO            |
> |-----------------|----------|---------|-----------------|----------------|
> |halfcheetah-jump | -1022.6  | 1808.6  | 5308.7$\pm$575.5| 4053.4$\pm$67.0|
> |ant-angle        | 866.7    | 2311.9  | 2776.9$\pm$43.6 | 809.2$\pm$73.0 |
> |sawyer-door-close| 5%       | 100%    | 98.3$\pm$3.0%   | 66.3$\pm$37.5% |
>
> As shown in the table above, the performance of COMBO is stable across random seeds and consistently outperforms the second-best approach, MOPO. We will run six random seeds of COMBO and other baselines on all the other domains and report 95%-confidence intervals in the final version of the paper.
>
> We address the additional concerns as follows.
>
> **"COMBO is still dependent on the sub-optimality of the data (offline + synthetic)."**: We acknowledge that COMBO does not attain the best performance on all domains and all dataset compositions; however, it does attain superior performance on the majority of the domains (all of the tasks that require generalization, image-based tasks, and 9 out of 12 D4RL tasks). COMBO is not the best performing method on some domains, and we have added this as a limitation to the discussion section of the paper. However, we would note that several celebrated papers in deep RL such as Rainbow DQN, QR-DQN, etc do not outperform the best baseline in all the tasks individually, but perform better in aggregate. Thus, we believe that the strong performance on most tasks and the theoretical analysis make the paper valuable to the community.
>
> **"It is not adequately explained how they have normalized return values of COMBO and other methods neither in the paper nor in the appendix."**: We use the normalization strategy recommended in the official D4RL protocol [1], and take the random scores and expert scores directly from D4RL open-sourced code (https://github.com/rail-berkeley/d4rl). We will add this clarification to the final version of the paper and will update the appendix with unnormalized return values for reference. This is the standard way to report D4RL results recommended in the original D4RL paper and used in most prior works, not something we came up with.
>
> **"I would invite the authors to release their code."**: We will open-source the code and add reference to it in the final version of the paper.

---

### Author Response · Authors · 2021-08-10
**Summary of changes**

We thank the reviewers for their feedback and a generally positive assessment of the paper. In this summary note, we would like to highlight our responses to the main concerns raised by the reviewers and the main experiments that we have added in the rebuttal period.

1. **[Reviewer GPHd, nBvp]** We have now run more seeds on the tasks that require generalization to unseen behaviors, totaling to **6 seeds** for COMBO and MOPO, which is the second best-performing method on these generalization tasks. COMBO still consistently outperforms MOPO on the generalization tasks. We will include more seeds for all methods on all the domains in the final version of the paper. Exact performance numbers can be found in the responses to **Reviewers GPHd** and **nBvp**.
2. **[Reviewer 4HWY, CcvH]** We have clarified that we tune our hyperparameters based on an offline hyperparameter selection scheme without requiring any rollouts in the real environment. Our offline hyperparameter selection scheme picks the hyperparameter based on the regularization objective of the Q-values. Details of this scheme can be found in the response to **Reviewers 4HWY** and **CcvH**.
3. **[Reviewer CcvH]** Regarding the comment that there is no new algorithm proposed, we have now compared COMBO to a direct combination of CQL and MBPO in the tasks that require generalization. COMBO significantly outperforms this naive combination. Exact performance numbers and additional clarifications on novelty can be found in the responses to **Reviewer CcvH**.
4. **[Reviewer CcvH]** Per the reviewer’s suggestion, we perform an ablation study on the rollout policy $\mu(a|s)$ on the generalization tasks shown in the performance table in our response to **Reviewer CcvH**.

We would appreciate it if the reviewers can please take a look at these changes and let us know if they have any other concerns.

---

### Decision · Program_Chairs · 2021-09-27

**Decision:**

Accept (Poster)

**Comment:**

The reviewers all agreed that this work is promising. The authors also provided important clarifications in the response, including increasing the number of runs and clarifying that hyperparameter selection was actually done with an approach that they developed for this paper suitable for the offline setting. This improvements undoubtedly make the paper stronger. However, the original omission of the strategy to select hyperparameters offline is a large omission, as it is a critical part of the algorithm. This algorithm should be discussed and contrasted to other approaches.

The theory was not extensively discussed (unfortunately), and relies heavily on the CQL theory. The CQL theory already has some constants that make for relatively loose bounds. Here, there are additional issues with the introduction of beta. The first main result relies on having a potentially very large beta, especially if you look at the proof and see the terms that beta*(small positive constant) is overcoming. The second result, on policy improvement, relies again a potentially large beta and on a term being positive that is not guaranteed to be positive (there is simply a discussion in the paper on why it reasonably could be positive in settings of interest). Theory for offline RL is hard, and so progress should be acknowledged. But, at the same time, given the complexity of these results, it is important to more clearly explain the limitations and what this theory can truly guarantee.

As a note, the concern about per-environment hyperparameter tuning was resolved in the discussion. It is reasonable to pick hyperparameters per environment, since you have an automated algorithm to do so. This is very different from optimizing (tuning) hyperparameters with sweeps, which is not an algorithm to be used in practice but rather an approach to evaluate methods.

As it stands, this work is borderline, and would highly benefit from another round to incorporate these changes and be re-reviewed. Some benefit of the doubt can be given that these changes may be done for the final paper, and so if there is space in the program, this paper could be accepted.